# Compliance towards WHO recommendations on antenatal care for a positive pregnancy experience: Timeliness and adequacy of antenatal care visit in Sub-Saharan African countries: Evidence from the most recent standard Demographic Health Survey data

Aklilu Habte[1]*, Aiggan Tamene[1], Tamirat Melis[2]

1 School of Public Health, College of Medicine and Health Sciences, Wachemo University, Hosanna, Ethiopia, 2 Department of Public Health, College of Medicine and Health Sciences, Wolkite University, Wolkite, Ethiopia

* akliluhabte57@gmail.com

**Data Availability Statement:** The data supporting the findings of this study can be obtained in

## Abstract

### Background

Timely and adequate antenatal care (ANC) visits are known to reduce maternal mortality by 20%. Despite the World Health Organization updating its recommendations from four to eight antenatal care contacts, data reporting in the SSA region focused primarily on four visits, and evidence on the timing and adequacy of ANC based on the current recommendation was limited. Hence, this study aimed at assessing the level of timely and adequate ANC visits and their determinants in the 18 Sub-Saharan African countries with the most recent DHS report (2016–2021).

### Methods

The data for this study were pooled from the most recent standardized Demographic and Health Survey data of sub-Saharan African countries from 2016–2021. A total of 171,183 (with a weighted frequency of 171,488) women were included and analyzed by using STATA version 16. To account for data clustering, a multivariable multilevel mixed-effect logistic regression analysis was run to determine the effects of each predictor on the receipt of timely and adequate ANC. Adjusted odds ratio with its corresponding 95% confidence interval was used to declare the statistical significance of the independent variables.

### Results

The receipt of timely and adequate antenatal care visits was 41.2% (95% CI: 40.9, 41.4) and 10.4% (95% CI: 9.9, 10.2), respectively. Wontedness of pregnancy [AOR = 1.18; 95% CI: 1.13, 1.24], being 1st birth order [AOR = 1.48; 95% CI: 1.41, 2.54], having a mobile phone [AOR = 1.49; 95% CI: 1.26, 2.32], and enrolled in Health insurance schemes [AOR =

anonymized form from the Demographic and Health Survey website at https://dhsprogram.com/Countries/ upon reasonable request in the same manner as the authors. The authors did not have any special access privileges that others would not have.

**Funding:** The author(s) received no specific funding for this work.

**Competing interests:** The authors have declared that no competing interests exist.

**Abbreviations:** AIC, Akaike's information criterion; ANC, Antenatal Care; AOR, Adjusted odds ratio; CSA, Central Statistical Agency; DHS, Demographic and health survey; EA, Enumeration area; EDHS, Ethiopian Demographic and Health Survey; ICC, Intra Class Correlation Coefficient; PCV, Proportional Change in Variance; PNC, Postnatal Care; PPP, Postpartum Period; SSA, Sub Saharan Africa.

2.03; 95% CI: 1.95, 2.42] were significantly associated with early initiation of ANC. Living in a lower community poverty level[AOR = 2.23; 95% CI: 1.90,2.66], being in the richest wealth quintile [AOR = 1.49; 95% CI: 1.36, 1.62], higher educational level [AOR = 3.63; 95% CI: 3.33, 3.96], the timing of ANC visit [AOR = 4.26; 95% CI: 4.08, 4.44], being autonomous in decision making [AOR = 2.29; 95% CI: 1.83, 2.54] and having a mobile phone [AOR = 1.89; 95% CI: 1.76, 2.52] were identified as significant predictors of adequate ANC uptake.

## Conclusion

The findings revealed a low coverage of timely and adequate ANC visits in SSA countries. Governments and healthcare managers in sub-Saharan African countries should leverage their efforts to prioritize and implement activities and interventions that increase women's autonomy, and economic capability, to improve their health-seeking behavior during pregnancy. More commitment is needed from governments to increase mobile phone distribution across countries, and then work on integrating mHealth into their health system. Finally, efforts should be made to increase the coverage of health insurance schemes enrolment for the citizens.

## Introduction

Maternal health refers to the health of women before pregnancy, all through pregnancy, intrapartum, and postpartum [1]. Sustainable Development Goal 3 (SDGs) emphasize maternal mortality reduction and improvements in women's health care, to reduce the global maternal mortality ratio (MMR) to 70 per 100,000 live births by 2030 [2, 3]. Despite progress and efforts, maternal and neonatal mortality and morbidity remain major public health concerns in low-income countries [4]. Low and middle-income countries(LMICs) accounted for 94% of all maternal deaths, of Sub-Saharan Africa (SSA) accounted for roughly two-thirds of all maternal deaths [4]. Despite having a very high MMR in 2017, this region has achieved a nearly 40% reduction in MMR since 2000 [5]. Complications during pregnancy, delivery, and the postpartum period are the leading causes of those death and disabilities among women of childbearing age in this region [6, 7].

Modifiable risk factors(undiagnosed infections, high blood pressure, diabetes), and other pre-existing health conditions frequently complicate pregnancy and endanger both mothers' and newborns' lives [8]. However, the majority of those adverse maternal and perinatal outcomes can be prevented by providing cost-effective interventions [6] like high-quality Antenatal care (ANC), which is defined by early initiation, and an adequate number of visits [9, 10]. ANC is one of the four pillars of WHO's safe motherhood initiative [11, 12]. Timely and regular ANC from a skilled provider reduces maternal mortality by 20% [13, 14]. It improves pregnancy outcomes by preventing, detecting, and treating danger signs early on through a series of scheduled visits [15]. It also serves as a gateway to maternal and child health services and provides an opportunity to organize the services required to ensure a healthy pregnancy, safe delivery, and a healthy mother-baby pair [16].

Early or timely initiation, accompanied by regular and adequate visits, has a significant impact on both maternal and fetal health [17]. The recent (2016) WHO recommendations on ANC for a positive pregnancy experience suggest a shift from a focused ANC model with a recommended minimum of four ANC visits (ANC4+) to a more expansive model emphasizing

contact number, timing, and content of care [18, 19]. This model recommends at least eight ANC contacts, with the first and eighth contacts occurring within the 12th and at 40th weeks of gestation, respectively [17, 18]. The increase from four to eight ANC contacts emphasizes the critical need of women who initiate ANC late, as they are less likely to meet the new recommended threshold for a positive pregnancy experience [20]. The first ANC occurring within the first trimester of pregnancy is described as the timely initiation of ANC [18, 21].

Although achieving the previously recommended ANC4+ has been a priority indicator for monitoring maternal health and has been widely reported [3, 22], timely first ANC is not widely reported or emphasized [20, 23]. The global coverage of at least one skilled ANC contact remains high at 86% [24], but the timely initiation of ANC is found to be low [20]. A recent study conducted in LMICs showed that 44.3% of all women had timely ANC initiation; 11.3% achieved ANC8+ [20]. Coverage of timely ANC initiation in low-income countries in 2013 was 24.0%, compared to 81.9% in high-income countries [21]. As per a DHS analysis of Uganda(2010) [25], and Ethiopia(2016) [26], 38.9% and 20.0% of women had a timely first ANC visit, respectively. Regarding the adequacy of ANC visits, analysis of Demographic and Health Survey (DHS) data from eight SSA countries found that only 7.7% of women complied with the new recommendation of at least eight visits [27]. Analysis of DHS data of Bangladesh [28] and Nigeria [29] showed only 6.0 and 17.4% of women made at least eight contacts, respectively. Financial constraints, distance to health facilities, cultural and religious beliefs around disclosure of pregnancy status, gender norms, lack of awareness of pregnancy signs and antenatal care schedules, pregnancy wantedness, perceptions on the need to start ANC early, and quality of care received were factors known to influence the timing and adequacy of ANC [20, 27, 28, 30–33].

Despite the World Health Organization (WHO) updating its recommendations from four to eight antenatal care contacts, data reporting in the SSA region focused primarily on four visits, and evidence on the timing and adequacy of ANC based on the current recommendation was limited [24]. As a result, more research is needed to characterize the number of women who commence ANC on time, the level of receiving recommended visits, the regions that are falling behind, and what it will take to improve the service uptake to meet the revised WHO guidelines. Hence, this study aimed at assessing the level of timely and adequate ANC visits and their determinants in the 18 SSA countries with the most recent DHS report(2016–2021). The findings of this study will be useful to service providers, policymakers, and programmers in their efforts to reduce maternal and child mortality by developing effective interventions to improve the quality of ANC.

## Methods

### Data source, population, and study period

This study was based on the most recent standardized DHS data from eighteen SSA countries. The DHS is a national survey that is carried out in 90 countries to collect data on basic health indicators. Data for each country were stored in men (MR), women (IR), children (KR), births (BR), and households (HR) files, and data for this study were obtained from the women (IR) file. The selected IR file contains information gathered about pregnancy, delivery, postnatal care, immunization, health, and nutrition. During the data inclusion process, the following criteria were considered: countries with recent standardized DHS reports conducted from 2016 to 2020, and countries with complete cases on the variables of interest (Table 1). Data of the study participants were accessed on November 23, 2022 from DHS office with a reasonable request. A total of 171,488 weighted women who had complete information on the uptake of at least one ANC visit were considered for the current study, and the entire analyses were

**Table 1. Description of the SSA countries included in the analysis with their respective sample size, 2016–2021.**

| Regions | DHS Year | Weighted sample size [n(%)] | Timely ANC [n(%)] | Adequate ANC [n(%)] |
|---|---|---|---|---|
| **Central Region** | **2016–2018** | **18,229(10.6)** | **8,723(47.8)** | **1,654(9.1)** |
| Angola | 2016 | 6,859(4.00) | 3,360(49.0) | 646(9.4) |
| Cameroon | 2018 | 11,370(6.6) | 5,364(47.2) | 1,008(5.9) |
| **Eastern Region** | **2016–2021** | **70,851(41.4)** | **29,443(41.6)** | **1,498(2.1)** |
| Burundi | 2017 | 8,827(5.1) | 4,205(47.6) | 42(0.5) |
| Ethiopia | 2016 | 4,757(2.8) | 1,540(32.4) | 156(3.3) |
| Madagascar | 2021 | 8,194(4.8) | 2,843(34.7) | 199(2.4) |
| Malawi | 2017 | 13,172(7.7) | 4,575(44.2) | 191(1.4) |
| Rwanda | 2020 | 6,158(3.6) | 3,695(60.0) | 16(0.3) |
| Uganda | 2016 | 9,902(5.8) | 2,926(29.5) | 187(1.9) |
| Zambia | 2018 | 7,182(4.2) | 2,654(37.0) | 91(1.3) |
| Mauritania | 2020 | 12,659(7.4) | 8,377(66.2) | 191(1.4) |
| **Western Region** | **2018–2020** | **79,645(46.4)** | **31,059(39.0)** | **14,191(17.8)** |
| Benin | 2018 | 7,814(4.5) | 4,398(56.3) | 809(10.4) |
| Gambia | 2020 | 10,611(6.2) | 4,552(42.9) | 444(4.2) |
| Guinea | 2018 | 4,548(2.6) | 1,470(32.3) | 172(3.8) |
| Liberia | 2021 | 7,768(4.5) | 5,605(72.1) | 2,101(27.0) |
| Mali | 2018 | 10,348(6.1) | 4,574(44.2) | 442(4.3) |
| Nigeria | 2018 | 32,228(18.8) | 7,662(23.8) | 8,612(26.7) |
| Sierra Leone | 2020 | 6,327(3.7) | 2,796(44.2) | 1,611(25.4) |
| **Southern region** | **2016** | **2,763(1.6)** | **1,387(50.2)** | 519(18.8) |
| South Africa | 2016 | 2,763(1.6) | 1,387(50.2) | 519(18.8) |
| **Total** | **2016–2021** | **171,488** | **70,613(41.2)** | **17,863(10.4)** |

conducted on them. Due to a lack of information on service uptake, a total of 10,971 respondents were excluded from the analysis (Missing values).

## Data collection tool and procedure

The data was collected through face-to-face interviews by trained data collectors using structured questionnaires developed in the official language of each country. A two-stage cluster sampling approach was used to choose respondents for the DHS survey. The initial phase was selecting sample locations (clusters) comprised of enumeration areas (EAs). The second stage was a systematic sample of houses. A household listing operation was done in each of the designated EAs, and households to be included in the survey were selected at random from the list. Finally, this study considered combined DHS data from 18 SSA countries, with a weighted sample of 171,488 reproductive-age women who had at least one child within five years preceding the survey. The detailed sampling technique used during DHS is discussed in the Demographic and Health Survey Sampling and Household Listing Manual developed by ICF International [34].

## Measurement of variables of the study

**Outcome variables:** the study has two outcome variables:

The first outcome variable: Was the timing of ANC commencement which was assessed by the question "When was your first antenatal check (in a month)?" the responses were converted into binary outcomes, as early and Late initiation. Women who received their first ANC

visit within the first 3 months (or 12 weeks after the onset of pregnancy) were categorized as "Early = 1", otherwise "Late = 0" [18].

The second outcome variable: Was the uptake of adequate ANC visits which was assessed by the question "How many times did you receive ANC visits throughout your last pregnancy?' The responses ranged from 1–20 visits and were then dichotomized into "Adequate = 1" (if a mother got at eight and more visits) and "Inadequate = 0" "if the mother got fewer than eight visits" [18].

**Explanatory variables.** Based on a review of the recent literature, variables with theoretical and practical significance in ANC uptake were extracted from the dataset and categorized into individual and community-level factors (Table 2).

**Table 2. List of the individual- and community-level factors that affect the timing and adequacy of ANC, in SSA countries 2016–2021.**

| Individual-level factors | |
|---|---|
| variables | Description |
| Maternal age | 15–24, 25–34, 35 and above |
| Marital status | In marital relations, live with a partner, in non-marital relation |
| Women's and husbands' education | No education*, primary, secondary and higher education |
| Occupation | Employed and unemployed* |
| Wealth index | Richest, richer, middle, porer, poorest* |
| Family size | ≤5memeber, >5 memeber* |
| Sex of head of Household | Female, male* |
| Parity | Nulliparous, primiparous, multiparous, Grand multiparous* |
| Birth order | 1, 2, 3, 4 and ≥5* |
| Pregnancy status | Wanted then, wanted later, wanted no more* |
| Termination of pregnancy | Yes, no* |
| Contraceptive utilization | Yes, no* |
| Decision-making capacity[a] | Autonomous, non-autonomous* |
| Frequency of reading a newspaper, listening to the radio, and watching television | Almost every day, at least once a week, less than once a week, not at all* |
| Distance to a health facility | Not a big problem, big problem* |
| Getting permission to go to a health facility for medical care | Not a big problem, big problem* |
| Getting money needed for treatment | Not a big problem, big problem* |
| Having a mobile phone | Yes, no* |
| Being member of health insurance schemes | Yes, no* |
| **Community-level factors** | |
| Residence | Urban, rural* |
| Region | Southern, Central, Western, Eastern* |
| Community level poverty | Low, moderate, high* |
| | *Reference for the categories |

[a] **Autonomy in decision-making:** Was measured using three constructs: (i) health care (healthcare autonomy), (ii.) major household purchases (economic autonomy), and (iii.) visits to friends or family (movement autonomy). A score of one (= 1) was assigned if women could decide alone or in collaboration with their husbands, whereas a score of zero (= 0) was assigned if a husband's decision was made alone or in collaboration with others. After all, a composite index was created and had a minimum and maximum value of 0 and 3, respectively. Women who scored at least one were considered autonomous; otherwise, they were considered non-autonomous [35, 36].

## Data management and statistical analyses

Compiled DHS reports of 18 SSA countries were checked for consistency and missing values. Cleaning, recoding, variable generation, labeling, and analysis were done by using STATA/SE version 16.0. The data were weighted to restore the statistical representativeness of the survey and to account for sample design. As a result, sample weights were used to estimate proportions and frequencies to account for disproportionate sampling and non-response. The weighting procedure was thoroughly discussed in the Demographic and Health Survey Sampling and Household Listing Manual developed by ICF International [34]. To describe the background characteristics of the study participants, descriptive statistics such as frequencies, percentages, and means were computed. The percentage and number of women in each category who received timely and adequate ANC were presented. To determine the distribution of timing and ANC adequacy across covariates, a cross-tabulation test was performed. The variance inflation factors (VIF) showed no collinearity between explanatory variables (VIF = 1.42).

The hierarchical nature of the DHS data contradicts the assumptions of the classic logistic regression model: independence of observations and homogeneity of variance. This violation occurred because women were nested within a cluster on the notion that women in the same cluster are more similar. Besides, unobserved cluster characteristics such as the availability and accessibility of health services, cultural norms, and dominant health beliefs may influence the receipt of timely and adequate ANC [34]. Thus, advanced models are required to overcome the shortcomings caused by inter-cluster heterogeneity, and a multilevel mixed-effects logistic regression model was preferred to identify significant predictors.

First, a multilevel bivariable logistic regression was performed to investigate the independent relationship between the covariates and the outcomes of interest (timing and adequacy of ANC visits). Those significant variables at p-values <0.25 were included in the multilevel mixed-effect multivariable logistic regression. Four models were used in this process: Models I, II, III, and IV. A model-I is a null model devoid of explanatory variables that shows the variance in timing and adequacy of ANC visits that could be ascribed to the cluster. Model II and III were models with the individual- and community-level factors, respectively. The full model (Model IV) examines the effect of individual and community-level predictors on both outcome variables concurrently. To demonstrate the strength and significance of the association, the adjusted odds ratio with the corresponding 95% confidence interval was calculated and reported. Variables with a p-value< 0.05 were considered to have a significant association with the outcome variables (timing and adequacy of ANC).

**Fixed and random effect estimates and model selection.**   Fixed and random effects were used to denote the measure of variation in the outcome of interest. In the random effect analysis, differences between clusters were examined using the intraclass correlation coefficient (ICC), a proportional change in variance (PCV), and median odds ratio (MOR). ICC is the proportion of variance explained by the population's group structure, whereas PCV is the total variation attributed to individual-level and community-level factors when compared to the null model [29, 37]. MOR is defined as the median value of the odds ratio between the clusters at high and low risk of early and adequate ANC visit when two clusters are randomly selected [37]. Of all the four models, the fourth one had the lowest AIC value in both cases (1st and 2nd outcome variables) and was selected as the best model fit for the data. Furthermore, as fitted models progressed from the empty model (Model-I) to Model-II, Model-III, and Model-IV, the value of the deviance (-2*log-likelihood) results consistently decreased, indicating that the fitted models were a better fit to the data.

**Ethical consideration and consent to participate.**   Following registration with possible justification, ICF International granted permission to access the dataset used for this study.

The retrieved data were only used for the registered research, and data were not shared with anyone other than the coresearchers. The information was kept private, and no attempt was made to identify any household or individual respondent. The DHS also declared that informed consent was obtained from all subjects and/or their legal guardian during the primary data collection.

## Results

### Sociodemographic characteristics of study participants

This study included 171,488 weighted women from 18 SSA countries who had at least one child within five years before the survey. The majority of study participants were from the western (79,645, 46.4%) and Eastern (70,851, 41.4%) %) regions of SSA. Nigeria, with 32,228 (18.8%), and South Africa, with 2,762(1.6%), were represented by the largest and smallest study participants, respectively. The mean (±SD) age of women in this study was 29.2(±7.24) years which, 43,856(25.6%) of them belong to the age group 25–34 years. one-third (33.3%) and 30.7% of women and their husbands, respectively, did not attend any formal education. The majority of the participants, 105,768(61.7) were from rural areas (Table 3).

### Obstetric characteristics of respondents

More than half of the respondents (51.0%) were multiparous (had 2–4 living children). Almost three-quarters (73.7%) of pregnancies were wanted, and 14.6% of women reported having faced pregnancy termination in their lifetime. Two-thirds (65.9%) of women received four or more antenatal care visits, and 5.8% gave birth by Caesarean section (Table 4).

### Health service-related characteristics of the respondents

Women had no exposure to reading newspapers, listening to the radio, and watching television in 85.9%, 43.4%, and 56.6% of cases, respectively. Nearly two-thirds, 109,430 (63.8%) of respondents were autonomous in decision-making. Long distances to a health facility, not having permission to go to a health facility, and not having enough money for treatment were major problems for 35.4%, 16.6%, and 50.1% of women, respectively. Only 7% of women were enrolled in health insurance schemes (Table 5).

### The overall uptake of timely ANC visits and its distribution across covariates

In eighteen SSA countries, the overall prevalence of timely initiation of ANC visits was 41.2% (95% CI: 40.9, 41.4). The central region had the highest proportion of timely ANC uptake (47.8%). When it comes to specific countries, Liberia and Nigeria had the highest (72.2%) and lowest (23.8%) prevalence, respectively (Fig 1).

Half (50.5%) of women in the richest wealth quintile started ANC within the first trimester as compared to 37.7% of women in the poorest wealth quintile group ($X^2$ = 1013.31, p<0.001). There was also a significant difference in early ANC uptake across women's educational level, with 52.0% of women who attend higher education starting their ANC visit on time, compared to 37.9% of those with no formal education($X^2$ = 644.30, p<0.001)(Table 3). The uptake of timely ANC initiation also varied across various obstetric characteristics of respondents. It was higher (51.0%) among women who received four or more ANC visits compared to women who received only one visit throughout their pregnancy (4.7%) ($X^2$ = 1013.31, p<0.001). It also varies across birth order, mode of delivery, and parity (Table 4). The majority of women who read newspapers, listen to the radio and watch television almost every day started their

**Table 3. Distribution of Sociodemographic characteristics of study participants across timeliness and adequacy of ANC visits in SSA, 2016–2021.**

| Variable categories | Total | Early initiation of ANC visits | | Adequacy of ANC visits ($>= 8$ visits) | |
|---|---|---|---|---|---|
| | | Yes [Frequency (%)] | Significance ($X^2$) | Yes [Frequency (%)] | Significance ($X^2$) |
| **Regions** | | | | | |
| Eastern | 70,851(41.4) | 29,443(41.6) | 270.97** | 1,498(2.1) | 1313.2** |
| Western | 79,646(46.4) | 31,059(39.0) | | 14,191(17.8) | |
| Central | 18,229(10.6) | 8,724(47.8) | | 1,655(9.1) | |
| Southern | 2,790(1.6) | 1,387(50.2) | | 519(18.8) | |
| **Current Age** | | 1 | | | |
| 15–24 | 50,021(29.1) | 19,906(39.8) | 59.38* | 3,959(7.9) | 387.68** |
| 25–34 | 79,166(46.2) | 33,386(42.2) | | 9,047(11.4) | |
| 35–49 | 42,301(24.7) | 17,322 (41.0) | | 4,857(11.5) | |
| **Marital status** | | | | | |
| In marital relation | 124,066(72.4) | 49,372(39.8) | 207.19** | 12,906(10.4) | 3.88 |
| Live with partner | 22,486(13.1) | 10,449(46.5) | | 2,438(10.8) | |
| Non-marital relation | 24,936(14.5) | 10,792(43.3) | | 2,518(10.1) | |
| **Educational status** | | | | | |
| No education | 57,054(33.3) | 21,641(37.9) | 644.30** | 3,476(6.1) | 612.31** |
| Primary | 56,282(32.8) | 22,770(40.5) | | 3,349(5.9) | |
| Secondary | 49,423 (28.8) | 21,663(43.8) | | 8,113(16.4) | |
| Higher | 8,729 (5.1) | 4,540(52.0) | | 2,924(33.5) | |
| **Husband's education** | | | | | |
| No education | 44,971(30.7) | 17,646 (39.2) | 517.42** | 2,618(5.8) | 746.53** |
| Primary | 38,920(26.6) | 14,717(37.8) | | 2,060(5.3) | |
| Secondary | 42,287(28.9) | 17,603(41.6) | | 6,648(15.7) | |
| Higher | 13,652(9.3) | 6,406(46.9) | | 3,503(25.7) | |
| Don't know | 6,694(4.6) | 3,437(51.3) | | 508(7.6) | |
| **Residence** | | | | | |
| Urban | 65,720(38.3) | 29,841(45.4) | 561.60** | 11,403(17.3) | 1623.14** |
| Rural | 105,768(61.7) | 40,772(38.6) | | 6,459(6.1) | |
| **Family size** | | | | | |
| ≤5memeber | 75,046(43.8) | 31,973 (42.6) | 52.40* | 9,407(12.5) | 467.36** |
| >5 memeber | 96,442(56.2) | 38,641(40.1) | | 8,455(8.8) | |
| **Wealth index combined** | | | | | |
| Poorest | 32,410(18.90) | 12,227(37.7) | 1013.31** | 1,599(4.9) | 1916.41** |
| Poorer | 35,001(20.41) | 13,213(37.8) | | 2,144(6.1) | |
| Middle | 35,552(20.73) | 13,622 (38.3) | | 3,091(8.7) | |
| Richer | 35,364(20.62) | 14,798(41.8) | | 4,411(12.5) | |
| Richest | 33,161(19.34) | 16,754(50.5) | | 6,619(20.0) | |
| Sex of household head | | | | | |
| Male | 134,920(78.7) | 53,876(39.9) | 305.02** | 14,000(10.4) | 16.84 |
| Female | 36,568(21.3) | 16,737 (45.8) | 0.00 | 3,862(10.6) | |
| Community level poverty | | | | | |
| Low | 60,363(34.9) | 24,614(40.8) | 20.54 | 8,645(50.1) | 1321.78 |
| Moderate | 58,057(34.1) | 24,067 (41.4) | | 4,429(25.7) | |
| High | 53,068(31.0) | 21,932(41.2) | | 4,188(24.3) | |

** p-values<0.001

* p-values<0.05

**Table 4. Distribution of obstetric characteristics of study participants across timeliness and adequacy of ANC visits in SSA, 2016–2021.**

| Variable categories | Total | Early initiation of ANC visits | | Adequacy of ANC visits ($\geq 8$ visits) | |
|---|---|---|---|---|---|
| | | Yes [Frequency (%)] | Significance ($X^2$) | Yes [Frequency (%)] | Significance($X^2$) |
| **Parity** | | | | | |
| Nulliparous | 1,682(1.0) | 691 (41.0) | 446.04** | 167(9.9) | 397.59** |
| Primiparous | 39,382(23.0) | | | 4,554(11.6) | |
| Multiparous | 87,580(51.0) | 36,819 (42.0) | | 10,085(11.5) | |
| Grand multiparous | 42,844 (25.0) | 15,680(36.6) | | 3,056(7.1) | |
| Caesarean delivery | | | | | |
| Yes | 161,634(94.2) | 65,173(55.2) | 767.85** | 16,087(9.9) | 494.41** |
| No | 9,854(5.8) | 5,4410 (40.3) | | 1,775(18.0) | |
| **Pregnancy status when she became pregnant** | | | | | |
| Wanted | 126,425(73.7) | 53,165(42.0) | 184.72** | 13,949(11.0) | 121.56* |
| Mistimed | 34,437(20.0) | 13,529(39.3) | | 2,995(8.7) | |
| Unwanted | 10,626(6.2) | 3,919(36.9) | | 918(8.6) | |
| Ever faced termination of pregnancy | | | | | |
| No | 146,455(85.4) | 59,650(40.7) | 76.83* | 14,941(10.2) | 27.50 |
| Yes | 25,033(14.6) | 10,963(43.8) | | 2,921(11.7) | |
| **Contraceptive utilization** | | | | | |
| Modern method | 49,072 (28.7) | 21,297() | 199.11* | 4,771(9.7) | 807.5** |
| Traditional method | 5,223(3.0) | 2,240 () | | 1,283(24.6) | |
| Nonusers | 117,193(68.3) | 47,076() | | 11,808(10.1) | |
| **Frequency of ANC** | | | | | |
| 1 visit | 7,966(4.7) | 2,232(28.0) | 712.13** | | |
| 2 visits | 14,323(8.4) | 2,645(18.5) | | | |
| 3 visits | 36,120(21.0) | 8,123 (22.5) | | | |
| $\geq$4 visits | 113,079(65.9) | 57,613(51.0) | | | |
| Timing of ANC initiation | | | | 11,153(64.6) | 1937.34** |
| Within 12 weeks | | | | 6,109(35.4) | |
| >12 weeks | | | | | |
| **Birth order** | | | | | |
| First | 36,948(21.6) | 16,492(45.3) | 571.51** | 4,246(11.5) | 54.20* |
| Second | 33,069(19.3) | 14,460(44.6) | | 4,141(12.5) | |
| Third | 27,634(16.1) | 11,720 (43.5) | | 3,254(11.8) | |
| Ourth | 21,870(12.8) | 8,770 (41.1) | | 2,230(10.2) | |
| Fifth and above | 51,966(30.3) | 19,171(38.3) | | 3,991(7.7) | |

** p-values<0.001

* p-values<0.05

ANC visits for their recent pregnancy on time as compared to their counterparts (p<0.001) (Table 5).

## Adequacy of ANC visit and its distribution across covariates

The overall uptake of adequate ANC (at least eight visits) was 10.4% (95% CI: 9.9, 10.2). The western region had the highest proportion of women receiving adequate ANC (17.8%). In terms of specific countries, Nigeria and Rwanda had the highest (26.7%) and lowest (0.3%) prevalence, respectively (Fig 2).

**Table 5.  Distribution of Health service-related characteristics of study participants across Timely initiation of ANC and the uptake of recommended items of care in SSA, 2016–2021.**

| Variable categories | Total | Early initiation of ANC visits | | Adequacy of ANC visits($\geq$ = 8 visits) | |
|---|---|---|---|---|---|
| | | Yes [Frequency (%)] | Significance ($X^2$) | Yes [Frequency (%)] | Significance ($X^2$) |
| **Reading newspaper** | | | | | |
| Not at all | 147,321(85.9) | 58,885(40.0) | 458.22** | 13,860(9.4) | 924.05** |
| Less than once a week | 14,948(8.7) | 6,979 (46.7) | | 2,459(16.4) | |
| At least once a week | 8,808(5.14) | 4,503 (51.1) | | 1,470(16.7) | |
| Almost everyday | 411(0.2) | 247(60.1) | | 74(18.1) | |
| **Listening to a radio** | | | | | |
| Not at all | 74,454(43.4) | 29,161(39.2) | 253.839* | 6,086(8.2) | 140.61* |
| Less than once a week | 38,626(22.5) | 16,303(42.2) | | 4,749(12.3) | |
| At least once a week | 55,392(32.3) | 23,645(42.7) | | 6,823(12.3) | |
| Almost everyday | 3,017(1.8) | 1,505(49.9) | | 204(6.8) | |
| **Watching television** | | | | | |
| Not at all | 97,094(56.6) | 36,735(37.8) | 985.04* | 6,252(6.4) | 1233.3* |
| Less than once a week | 25,804(15.5) | 11,059(42.9) | | 3,354(13.0) | |
| At least once a week | 43,576(25.1) | 20,346(46.7) | | 7,849(18.0) | |
| Almost everyday | 5,013(2.9) | 2,473(49.3) | | 407(8.12) | |
| **Had a mobile phone** | | | | | |
| No | 80,091(46.7) | 28,890(36.0) | 1400.01* | 4,832(6.0) | 1721.23* |
| Yes | 91,398(53.3) | 41,724(46.7) | | 13,030(14.3) | |
| **Autonomy in decision making** | | | | | |
| Non-autonomous | 62,058(36.2) | 24,409(39.3) | 157.56* | 5,622(9.0) | 135.65* |
| Autonomous | 109,430(63.8) | 46,204 (42.2) | | 12,240(11.2) | |
| **Distance to a health facility** | | | | | |
| Big problem | 60,690(35.4) | 24,122(39.8) | 12.759 | 4,831(8.0) | 478.99* |
| Not a big problem | 110,798(64.6) | 46,492(42.0) | | 13,031(11.8) | |
| **Getting permission to go to a health facility** | | | | | |
| Big problem | 28,452(16.6) | 12,152(42.7) | 50.07 | 2,6789(9.4) | 38.17 |
| Not a big problem | 143,037(83.4) | 58,462 (40.9) | | 15,184(10.6) | |
| **Getting money needed for treatment** | | | | | |
| Big problem | 85,816(50.1) | 34,221(39.9) | 160.65* | 7,698(9.0) | 321.85* |
| Not a big problem | 85,672(49.9) | 36,392(42.5) | | 10,164(11.9) | |
| **Covered by health insurance** | | | | | |
| No | 159,301(92.9) | 63,467(39.8) | 1230.31* | 16,872(10.4) | 97.33 |
| Yes | 12,187(7.1) | 7,146 (58.6) | | 991(8.1) | |

** p-values<0.001

* p-values<0.05

In the richest wealth quintile, 20.5% of women received adequate ANC, compared to 4.9% of women in the poorest wealth quintile ($X^2$ = 1916.41, p<0.001). There was also a significant difference in adequate ANC uptake across women's educational levels, with 33.5% of women in higher education getting adequate ANC visits, compared to 6.19% of those with no formal education ($X^2$ = 612.31, p<0.001) (Table 3). The uptake of adequate ANC visits varied across the obstetric characteristics of respondents. It was higher (64.6%) among women who started their visit within 12 weeks of gestation compared to women who started later ($X^2$ = 1937.34, p<0.001). It also varies by birth order, mode of delivery, and parity (Table 4). The magnitude

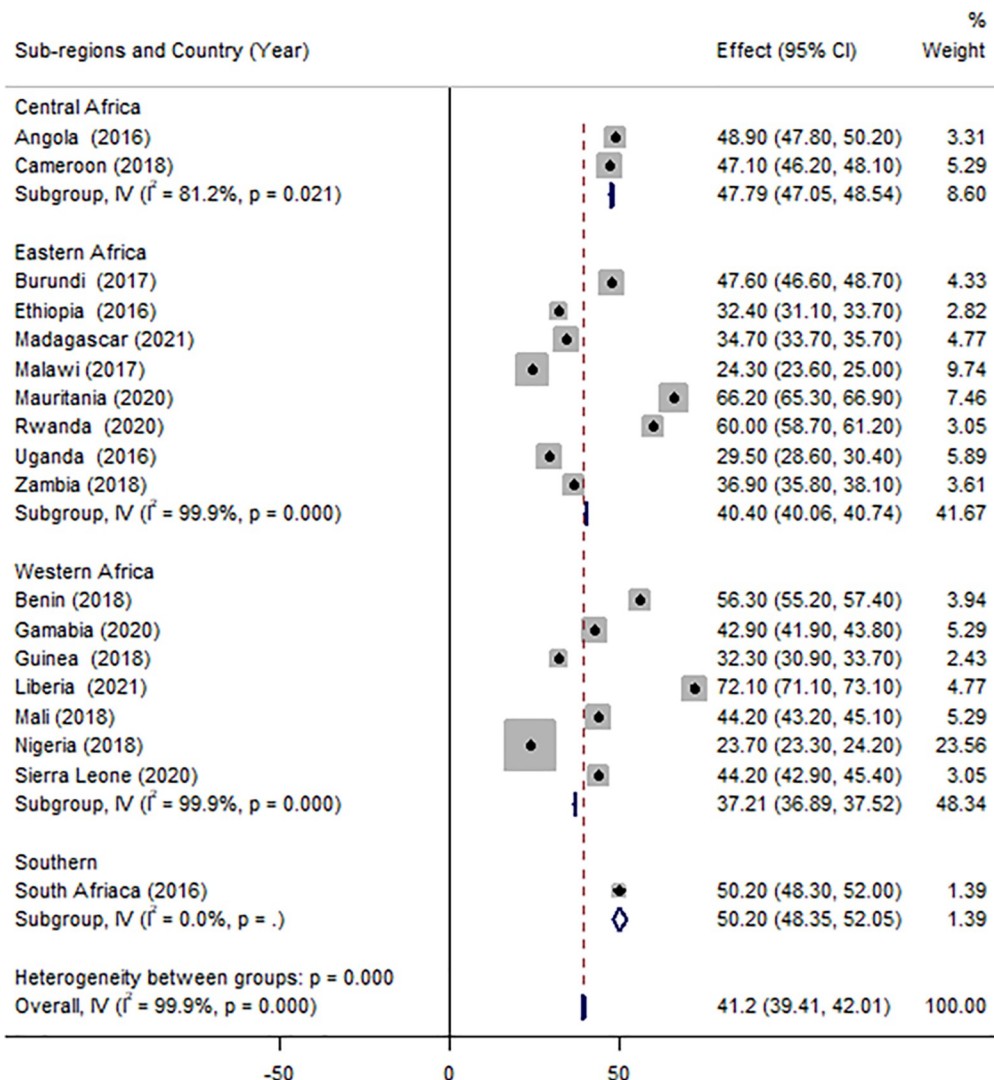

**Fig 1. Forest plot depicting the pooled prevalence of early ANC initiation by country and sub-regions, 2016–2021.**

of receiving adequate ANC varies among women who read newspapers, listen to the radio, and watch television (p<0.001) (Table 5).

## The receipt of WHO-recommended items of care during visits

Data on the provision of the essential components of ANC services during pregnancy were considered: weight and blood pressure measurement, urine and blood sample testing, TT injection at least twice, iron supplementation, deworming for intestinal parasites, HIV/AIDS prevention counseling, and provision of HIV test as part of ANC service. When the individual components were considered, blood pressure measurement was the most common item received by 152,705 (89.0%) of women, closely followed by iron supplementation (88%) and a blood sample taken (87.6%). Ultrasound examination was a component received by only 9.1% of the women (Fig 3).

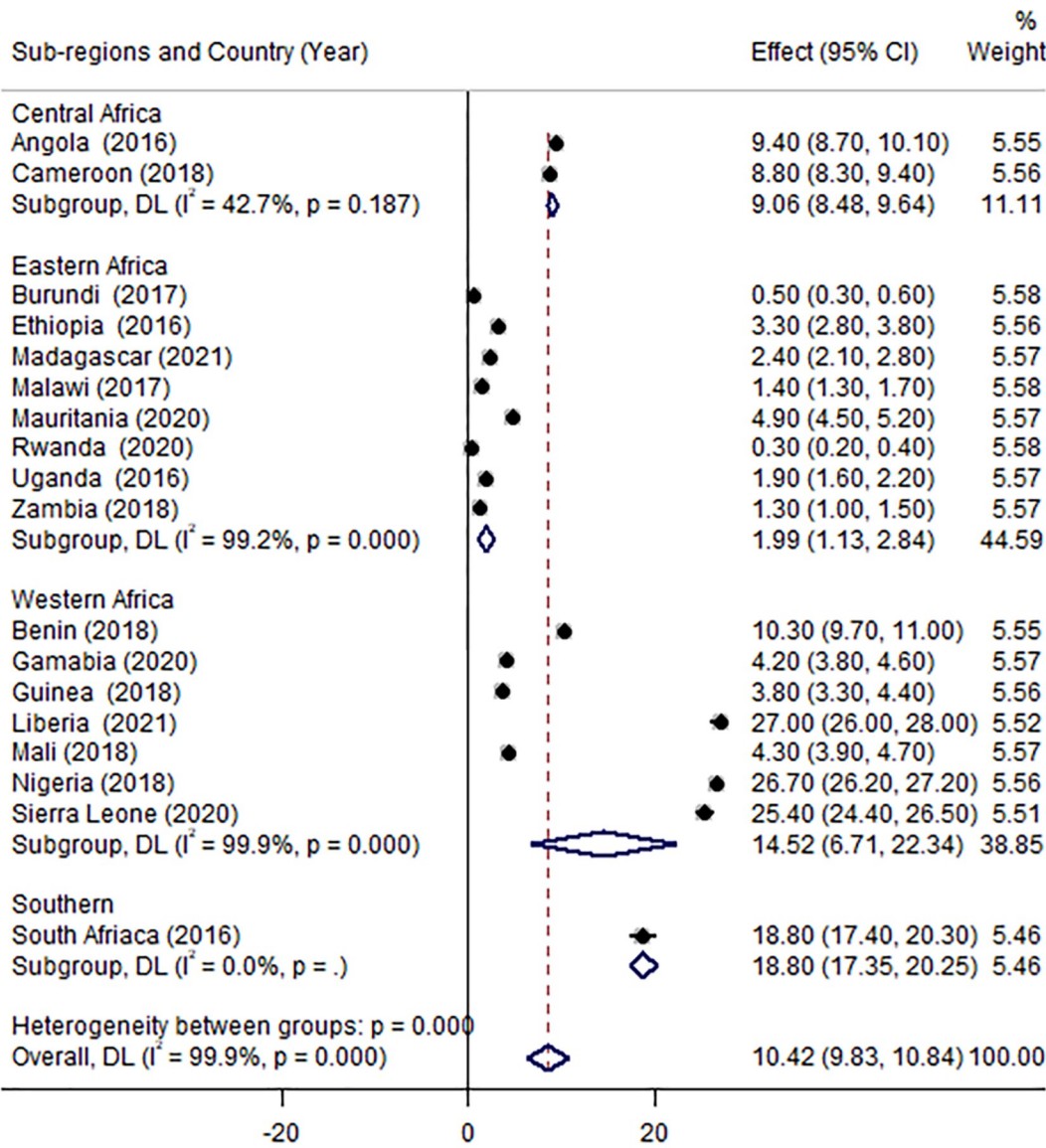

**Fig 2. Forest plot depicting the pooled prevalence of adequate ANC visits by country and sub-regions, 2016–2021.**
NOTE: Weights and between-subgroup heterogeneity test are from random-effects model.

### Determinants of early initiation of ANC service

In bivariable multilevel logistic regression, variables namely region, age of women, respondents' educational status, marital status, residence, family size, wealth index, the sex of head of household, wontedness of pregnancy, experiencing termination of pregnancy, contraceptive uptake, birth order, reading a newspaper, listening radio, watching television, having a mobile phone, autonomy in decision-making, having money for medical care, and enrollment in health insurance were associated with early initiation of ANC at p-value <0.25. The preceding multilevel bivariable logistic regression analysis presents the unadjusted effects of the explanatory variables on the timely initiation of ANC. Thus, a multilevel mixed-effect multivariable logistic regression analysis was performed to identify variables significantly associated with the

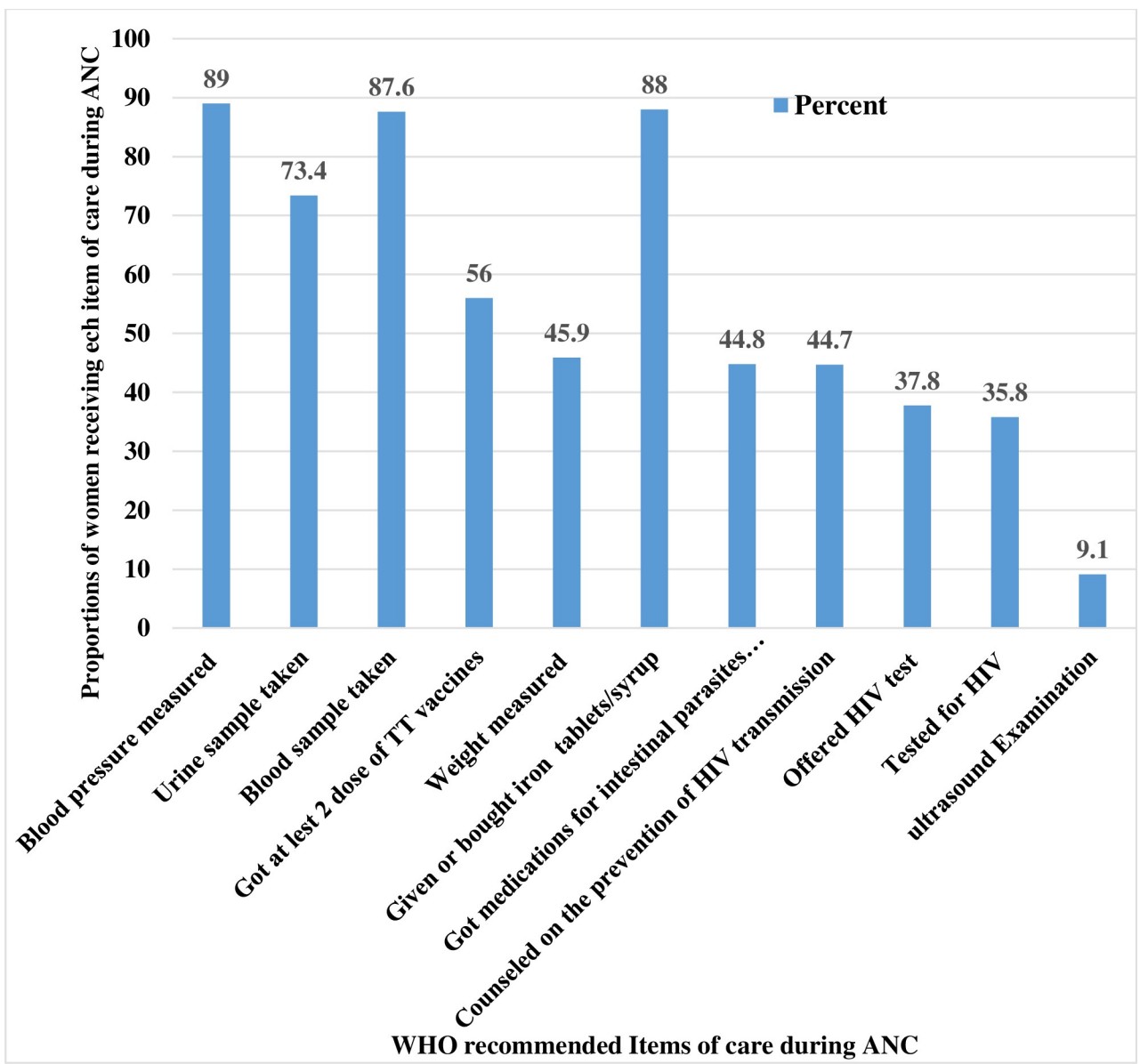

**Fig 3. Distribution of contents care provided to women during ANC visits in SSA countries, 2016–2021.**

outcome of interest. Accordingly, the region where the respondents lived, planning status of the pregnancy, birth order, having a phone number, and enrollment in health insurance schemes were found to be significantly associated with the early initiation of ANC visits.

Women from Eastern and Central SSA regions had a 41% (AOR = 1.41, 95% CI: 1.32, 1.48) and 24% (AOR: 1.24; 95% CI: 1.17, 1.32) higher chance of starting ANC at the right time, respectively. The pregnancy planning status was found to be a significant predictor of early initiation of ANC. When compared to their counterparts, the odds of starting ANC on time were 1.18 times higher in women with wanted pregnancies (AOR = 1.18, 95% CI: 1.13, 1.24). When compared to women with more than the fifth birth order, those with the first birth order had a 48% higher chance of initiating ANC visits on time (AOR = 1.48; 95% CI: 1.41, 1.54). Having a mobile phone was also identified as a significant predictor of the timely initiation of ANC.

Those women who had a mobile phone had a 49% higher chance of commencing their prenatal care on time (AOR = 1.49; 95% CI: 1.26, 1.82). The odds of starting ANC follow-up in the first trimester were 2 times higher among those women who enrolled in health insurance schemes (AOR = 2.03; 95% CI: 1.95, 2,42).

The ICC in the empty model implied that the variation across countries accounted for 15.3% of the total variation in the receipt of adequate ANC visits. Furthermore, the MOR was 1.35 (95% CI: 1.26, 1.43), indicating that when women moved from late ANC initiation countries, their odds of getting timely ANC visit increased by 1.35 times. Both individual- and community-level variables explained 14.3% of the national variation seen in the empty model (PCV = 14.3%) (Table 6).

## Determinants of adequate ANC visits among women

Again, a bivariable multilevel logistic regression was done between each explanatory variable and outcome variables. Accordingly, region, level of community poverty, age of women, respondents' educational status, residence, wealth index, the sex of the head of household, wontedness of pregnancy, experiencing termination of pregnancy, birth order, reading a newspaper, watching television, having a mobile phone, autonomy in decision-making, having money for medical care, ease of distance to a nearby health facility, autonomy in decision making and enrollment in health insurance were associated with the uptake of adequate ANC at p-value <0.25. Community-level poverty, the household wealth index, women's educational status, early initiation ANC visits, being autonomous in decision-making and having a mobile phone were identified as significant predictors of adequate ANC service uptake in a multilevel mixed-effect multivariable logistic regression analysis.

As compared to women from the eastern region, the odds of receiving adequate ANC visits were 13.1 times (AOR = 13.16, 95% CI: 12.34, 14.04) and 13.42 times (AOR = 13.42, 95% CI: 11.74, 15.33) higher for those women from western and southern regions, respectively. Women living in lower community level poverty were 2.23 times more likely to receive adequate ANC visits than women living in higher community level poverty (AOR = 2.23, 95% CI: 1.90, 2.66). Similarly, the odds of receiving adequate ANC visits were 1.49 times higher among those women in the richest wealth quintile as compared to their counterparts (AOR = 1.49, 95% CI: 1.36, 1.62). As compared to women with no formal education, the likelihood of receiving adequate ANC visits was 3,63 times higher for those who attended higher education (AOR = 3.63, 95% CI: 3.33, 3.96). Women who began their first ANC visit within 12 weeks of gestation had a 4.26-fold higher chance of receiving adequate ANC visits (AOR = 4.26, 95% CI: 4.08, 4.44). Those women who had a mobile phone had an 89% higher chance of receiving adequate ANC (AOR = 1.89; 95% CI: 1.76, 2.32). In terms of decision-making capabilities, autonomous women had a 2.29 times higher chance of receiving adequate ANC than non-autonomous women (AOR = 2.29; 95% CI: 1.83, 2.54).

The ICC in the empty model implied that the variation across countries accounted for 40.9% of the total variation in the receipt of adequate ANC visits. Furthermore, the MOR was 4.22 (95% CI:3.95, 4.49), indicating that when women moved from low to high ANC usage countries, their odds of getting adequate ANC increased by 4.22 times. Both individual- and community-level variables explained 39.5% of the national variation seen in the empty model (PCV = 39.5%) (Table 7).

## Discussion

Antenatal care coverage is a key performance indicator that has been reported globally to assess maternal health [20]. Although evidence on the number of ANC contacts is readily

**Table 6. Results of a multilevel mixed-effect multivariable logistic regression analysis to identify the factors affecting the early initiation of ANC in SSA, 2016–2021.**

| Variable categories | Model I(null model) | Model II (individual-level factors) | Model III (community-level factors) | Model-IV (full model) |
|---|---|---|---|---|
| | AOR (95% CI) | AOR (95% CI) | AOR (95% CI) | AOR(95% CI) |
| **Current Age** | | | | |
| 15–24 | | 1 | | |
| 25–34 | | 1.15(1.12, 1.19) | | 1.04(0.98,1.18) |
| 35–49 | | 1.20(1.15, 1.25) | | 1.19(0.83, 1.24) |
| **Educational status** | | | | |
| No education | | 1 | | 1 |
| Primary | | 0.95(0.92, 0.97) | | 0.83(0.79, 1.15) |
| Secondary | | 0.94(1.92,1.07) | | 0.98(0.76, 1.19) |
| Higher | | 1.24(0.90, 1.43) | | 1.12(0.85,1.21) |
| **Family size** | | | | |
| < = 5 member | | 1 | | 1 |
| >5 members | | 0.96(0.93, 0.98) | | 0.95(0.93, 0.97) |
| **Wealth index combined** | | | | |
| Poorest | | 1 | | 1 |
| Poorer | | 0.94(0.91, 0.97) | | 0.93(0.90, 0.96) |
| Middle | | 0.88(0.85, 0.91) | | 0.96(.84, 1.09) |
| Richer | | 0.92(0.89, 1.06) | | 1.01(0.96, 1.03) |
| Richest | | 1.13(1.08, 1.17) | | 1.15(0.98, 1.23) |
| **Sex of household head** | | | | |
| Male | | 1 | | 1 |
| Female | | 1.26(1.23, 1.29) | | 1.12(0.98, 1.23) |
| **Pregnancy status** | | | | |
| Wanted | | 1.33(1.08, 1.18) | | 1.18(1.13,1.24)* |
| Mistimed | | 1.04(0.99, 1.09) | | 1.07(0.98, 1.12) |
| Unwanted | | 1 | | 1 |
| **termination of pregnancy** | | | | |
| No | | 1 | | 1 |
| Yes | | 1.12(1.05,1.14) | | 1.07(0.97, 1.11) |
| **Contraceptive usage** | | | | |
| Modern method | | 1.08(1.06, 1.11) | | 1.04(0.95, 1.07) |
| Traditional method | | 0.97(0.91, 1.03) | | 0.95(0.89, 1.01) |
| Non-users | | 1 | | 1 |
| **Frequency of ANC** | | | | |
| 1 visit | | 1 | | 1 |
| 2 visits | | 1.33(0.59,0.68) | | 1.43(0.97, 1.68) |
| 3 visits | | 1.53(0.88, 1.68) | | 1.63(0.86, 1.88) |
| ≥4 visits | | 3.11(2.96, 3.28) | | 3.28(3.10, 3.45)* |
| **Birth order** | | | | |
| First | 1.52(1.46, 1.59) | | | 1.48(1.41, 1.54)* |
| Second | | 1.40(1.34,1.46) | | 1.37(1.32, 1.43)* |
| Third | | 1.26(1.21,1.31) | | 1.24(1.20, 1.29) |
| Fourth | | 1.11(1.07,1.15) | | 1.10(0.96, 1.14) |
| Fifth and above | | 1 | | 1 |
| **Listening to a radio** | | | | |
| Not at all | | 1 | | 1 |

*(Continued)*

**Table 6.** (Continued)

| Variable categories | Model I(null model) | Model II (individual-level factors) | Model III (community-level factors) | Model-IV (full model) |
|---|---|---|---|---|
| | AOR (95% CI) | AOR (95% CI) | AOR (95% CI) | AOR(95% CI) |
| < once a week | | 1.02(0.99, 1.05) | | 1.03(0.91, 1.07) |
| At least once a week | | 0.95(0.92, 0.97) | | 0.96(0.93, 1.28) |
| Almost everyday | | 1.06(0.97, 1.17) | | 1.12(1.02, 1.23) |
| **Watching TV** | | | | |
| Not at all | | 1 | | 1 |
| < once a week | | 1.05(1.02, 1.08) | | 1.07(0.95, 1.12) |
| At least once a week | | 1.07(1.04, 1.10) | | 1.09(0.92, 1.13) |
| Almost everyday | | 1.09(0.92, 1.17) | | 1.01(0.92, 1.09) |
| **Had a mobile phone** | | | | |
| No | | 1 | | 1 |
| Yes | | 1.27(1.24, 1.29) | | **1.49(1.26, 2.32)**[*] |
| **Decision making** | | | | |
| Non-autonomous | | 1 | | 1 |
| Autonomous | | 1.15(1.12, 1.17) | | 1.10(0.95, 1.14) |
| **Distance to a health facility** | | | | |
| Big problem | | 1 | | 1 |
| Not a big problem | | 1.14().91, 1.16) | | 1.12(1.09, 0.98) |
| **To Get money for treatment** | | | | |
| Big problem | | 1 | | 1 |
| Not a big problem | | 1.01(0.98,1.03) | | 1.04(0.91,1.13) |
| **Covered by health insurance** | | | | |
| No | | 1 | | 1 |
| Yes | | 2.22(2.13, 2.32) | | **2.03(1.95, 2.42)**[*] |
| **Regions** | | | | |
| Eastern | | | 0.98(0.77, 1.09) | 1.40(1.32, 1.48) |
| Western | | | 0.88(0.86, 1.00) | 0.99(0.94, 1.05) |
| Central | | | 1.02(0.99, 1.04) | 1.24(1.17, 1.32) |
| Southern | | | 1 | 1 |
| **Residence** | | | | |
| Urban | | | 1.13(1.08, 1.21) | 1.04(0.98, 1.07) |
| Rural | | | 1 | 1 |
| **Random effects** | | | | |
| Variance | 1.19 | 1.03 | 1.04 | 1.02 |
| ICC | 15.3% | 13.4% | 10.3% | 11.5 |
| AIC | 232,766.9 | 229,766.9 | 229,778.5 | 226,106.5 |
| BIC | 232,887.0 | 230,787.3 | 229,838.8 | 226,508.5 |
| MOR | 1.52(1.48, 1.56) | 1.41(1.37, 1.45) | 1.42(1.38, 1.46) | 1.35 (1.26, 1.43) |
| PCV | Reference | 13.4% | 12.6% | 14.3% |
| **Model fitness** | | | | |
| Log-likelihood | -115381.4 | -114,098.7 | -114,883.2 | -113, 013.2 |
| Deviance | 230,768.8 | 228,197.4 | 229,766.4 | 226, 026.4 |

**Key:** 1: Reference category; AOR = Adjusted odds ratio

** Statistically significant at p-value <0.05

**Table 7. Results of a multilevel mixed-effect multivariable logistic regression analysis to identify determinants of the receipt of adequate ANC in SSA, 2016–2020.**

| Variable categories | Model I(null model) | Model II (individual-level factors) | Model III (community-level factors) | Model-IV (full model) |
|---|---|---|---|---|
| | AOR (95% CI) | AOR (95% CI) | AOR (95% CI) | AOR(95% CI) |
| **Current Age** | | | | |
| 15–24 | | 1 | | |
| 25–34 | | 1.19(1.13,1.26) | | 1.04(0.98,1.18) |
| 35–49 | | 1.62(1.51, 1.73) | | 1.29(0.83, 1.54) |
| **Educational status** | | | | |
| No education | | 1 | | 1 |
| Primary | | 0.84(0.80, 1.29) | | 1.46(1.38, 1.54) |
| Secondary | | 1.90(1.80, 2.00) | | 2.11(1.99, 2.23) |
| Higher | | 3.80(3.50, 4.13) | | 3.63(3.33, 3.96)* |
| **Wealth index combined** | | | | |
| Poorest | | 1 | | 1 |
| Poorer | | 1.18(1.10, 1.27) | | 0.93(0.90, 0.96) |
| Middle | | 1.35(1.26, 1.44) | | 1.16(0.84, 1.09) |
| Richer | | 1.36(1.27, 1.46) | | 1.28(1.19, 1.388)* |
| Richest | | 1.43 (1.33,1.55) | | 1.49(1.36, 1.62)* |
| **Sex of household head** | | | | |
| Male | | 1 | | 1 |
| Female | | 1.04 (1.0, 1.09) | | 1.12(0.98, 1.23) |
| **Parity** | | | | |
| Grand multiparous | | 1 | | 1 |
| Multiparous | | 1.58(1.45, 1.72) | | 1.28(0.97, 1.51) |
| Primiparous | | 1.02(1.75, 2.33) | | 1.23(0.92, 1.38) |
| Nulliparous | | 1.94(1.5, 2.44) | | 1.27(0.98, 1.40) |
| **Pregnancy status** | | | | |
| Unwanted | | 1 | | |
| Mistimed | | 0.94(0.86, 1.03) | | 1.07(0.98, 1.12) |
| Wanted | | 0.99(0.92, 1.07) | | 1.18(1.13,1.24) |
| **Termination of pregnancy** | | | | |
| No | | 1 | | 1 |
| Yes | | 1.08(1.03, 1.13) | | 1.07(0.97, 1.11) |
| **Timing of ANC initiation** | | | | |
| Late(>12 wks) | | 1 | | 1 |
| Early (<12 wks) | | 3.6(3.45, 3.73) | | 4.26(4.08, 4.44)* |
| **Reading newspaper** | | | | |
| Not at all | | 1 | | 1 |
| < once a week | | 0.87(0.82, 0.93) | | 1.03(0.91, 1.07) |
| At least once a week | | 0.93(0.86, 1.04) | | 0.96(0.93, 1.28) |
| Almost everyday | | 1.25(0.86, 1.81) | | 1.12(0.92, 1.14) |
| **Watching television** | | | | |
| Not at all | | 1 | | 1 |
| < once a week | | 1.09(0.56, 0.74) | | 1.07(0.95, 1.12) |
| At least once a week | | 1.12(1.06, 1.18) | | 1.09(0.92, 1.13) |
| Almost everyday | | 1.17(1.11,1.24) | | 1.01(0.92, 1.09) |
| **Had mobile phone** | | | | |
| No | | 1 | | 1 |

*(Continued)*

**Table 7.** (Continued)

| Variable categories | Model I(null model) | Model II (individual-level factors) | Model III (community-level factors) | Model-IV (full model) |
|---|---|---|---|---|
| | AOR (95% CI) | AOR (95% CI) | AOR (95% CI) | AOR(95% CI) |
| Yes | | 1.61(1.24, 1.70) | | 1.89(1.76, 2,32)* |
| **Decision making** | | | | |
| Non-autonomous | | 1 | | 1 |
| Autonomous | | 2.35(1.81, 2.49) | | 2.29(1.83, 2.54)* |
| **Distance to a health facility** | | | | |
| Big problem | | 1 | | 1 |
| Not a big problem | | 1.07(1.02, 1.12) | | 1.12(1.09, 0.98) |
| **Accessing money for health service** | | | | |
| Big problem | | 1 | | 1 |
| Not a big problem | | 0.96(0.92, 1.01) | | 1.04(0.91,1.13) |
| **Covered by health insurance** | | | | |
| No | | 1 | | 1 |
| Yes | | 1.22(1.08, 1.52) | | 1.03(0.95, 2.42) |
| **Regions** | | | | |
| Eastern | | | 1 | 1 |
| Central | | | 5.03(4.65, 5.45) | 6.30(5.28, 6.30) |
| Southern | | | 14.24(12.70, 15.96) | 13.42(11.74, 15.34) |
| Western | | | 8.98(8.48, 9.53) | 13.16(12.34, 14.0) |
| **Residence** | | | | |
| Rural | | | 1 | 1 |
| Urban | | | 1.95(1.87, 2.03) | 1.29(0.98, 1.37) |
| **Community level poverty** | | | | |
| Higher | | | 1 | 1 |
| Moderate | | | 0.87(0.72, 1.05) | 0.81(0.67, 0.98) |
| Low | | | 2.65(2.26, 3.11) | 2.23(1.9, 2.66)* |
| **Random effects** | | | | |
| Variance | 2.28 | 1.97 | 1.35 | 1.38 |
| ICC | 40.9% | 37.5% | 27.9% | 27.8% |
| AIC | 99,532.0 | 90745.0 | 91023.8 | 82749.14 |
| BIC | 99,552.1 | 91096.8 | 91104.2 | 83161.21 |
| MOR | 4.22(3.95, 4.49) | 3.82(3.58, 4.06) | 2.93(2.78, 3.08) | 2,93(2.78, 3.08) |
| PCV | Reference | 13.6 | 44.3 | 39.5 |
| **Model fitness** | | | | |
| Log-likelihood | -49764.01 | -45,337.5 | -44,763.9 | -40,733.5 |
| Deviance | 99528.02 | 90,675 | 89,562.8 | 81,467.0 |

**Key:** 1: Reference category; AOR = Adjusted odds ratio

** Statistically significant at p-value <0.05 in the final model

available, the reports were based on the previously recommended ANC4+, and countries' performance based on the new WHO recommendation [18], particularly in SSAs, was not thoroughly assessed. Thus, effective ANC promotion and a positive pregnancy experience among women in the SSA region necessitate a thorough understanding of the timing and adequacy of ANC contacts. The current study was aimed at assessing the magnitude and predictors of timing and adequacy of ANC among women of SSA countries based on the most recent DHS data (2016–2021).

According to the current study, less than half (41.2%, 95% CI: 40.9, 41.4) of women in SSA countries initiated ANC during the first trimester of pregnancy, as recommended by WHO. This finding is lower than a recent study conducted in LMICs(44.3%) [20] and higher than DHS analyses of Uganda (38.9%) [25], and Ethiopia(20.0%) [26]. On the other hand, 10.4% (95% CI: 9.9, 10.2) of women in SSA got the newly set WHO recommendation of a minimum of 8 contacts which was lower than a study in LMICs(11.3%) [20], and DHS report of Nigeria (17.4%) [29] but higher than DHS analysis of eight SSA from 2010-2016(7.7%) [27], Bangladesh (6.0%) [28]. Overall, the current study reveals that compliance with the updated recommendation is low, most likely because it has not been adopted as a national policy in all countries, and more emphasis should be placed on its implementation.

Planning status of the pregnancy, birth order, having a phone number, and enrollment in health insurance schemes were found to be significantly associated with the early initiation of ANC visits. On the other hand, Community-level poverty, the household wealth index, women's educational status, early initiation ANC visits, being autonomous in decision-making, and having a mobile phone were identified as significant predictors of adequate ANC service uptake.

Birth order was identified as a factor significantly associated with the early initiation of ANC, with women with the first birth order being more likely to begin their third ANC visit earlier than mothers with birth orders of five or more. Further analysis of national studies conducted in the Brussels Metropolitan region [38], Haiti [39], Zambia [40], Nigeria [41], Uganda [42], and Ethiopia [10, 26, 43] supported this finding. The possible justification is that for women with first birth orders, the pregnancy is new to them, and they are concerned about having to deal with pregnancy complications, so they want to contact healthcare providers as soon as possible. Women with high birth orders, on the other hand, may be preoccupied with child-rearing and other home activities, which prevents them from seeking an early ANC visit [43]. As a result, women with high birth orders deserve special attention because they account for the highest proportion of the population (30.3%).

The odds of receiving early ANC is higher among those women with a wanted pregnancy. This finding was supported by studies conducted in Rwanda [25], Kenya [44], and Ethiopia [10, 26]. This could be because those women were well prepared from the start and had a better chance of detecting the pregnancy earlier at health facilities, leading them to begin the visits earlier. Besides, a woman with a wanted pregnancy is more likely to seek appropriate care for her pregnancy and is eager to obtain any information presumed to be important for their fetus from health care professionals and peers or friends, all of which leads them to begin their ANC early. Couples, on the other hand, are typically less vigilant about receiving ANC services in the event of unwanted pregnancy [28]. Thus, healthcare providers should be trained in identifying unwanted pregnancies and linking them to service delivery points by providing culturally appropriate support to those women [45].

The likelihood of early initiation of ANC visits was higher among those women who were enrolled in health insurance schemes. This was supported by studies conducted in Rwanda [25], Ghana [46, 47], Nigeria [48] and Taiwan [49]. A systematic review conducted in low- and middle-income countries also revealed that health insurance can influence the use and quality of maternal and child services and is known to potentially improve maternal and neonatal health outcomes [50]. This could be because health insurance schemes act as a pro-poor equalizing agent in healthcare financing by reducing out-of-pocket payments [51], which could offset the impact of monetary shortages that cause ANC to be delayed. Such health insurance schemes are known to improve the utilization of healthcare through financial protection, resource mobilization, and social exclusion [52]. Thus, the governments of SSA countries should work on the enrollment of their population in those schemes.

Having a mobile phone was also positively associated with the uptake of timely and adequate ANC visits. This finding was supported by studies conducted in Nigeria [53], Ghana [54], and Zanzibar [55]. This could be because women who owned a mobile phone were more likely to receive text message reminders for ANC appointments and other health information which resulted in positive health behavior, which could lead to them receiving timely and adequate ANC visits [56]. Thus, using mobile phones in health systems through mHealth to improve antenatal care service utilization is vital for SSA countries, which have the lowest antenatal care attendance compared to other regions [57].

The odds of receiving adequate ANC visits were higher among those women living in low community poverty levels and households with the richest wealth quintile as compared to their counterparts. Studies conducted in Bangladesh [28], Nepal [58], Nigeria [59], and Pakistan [23], supported this finding. This might be due to women in better socioeconomic status having more ability to pay for both direct and indirect health care costs. Economic factors may influence women's health-seeking behavior in many ways in the SSA region, where 40% of the population lives below the US$1.90-per-day poverty line in 2018, accounting for two-thirds of the global extremely poor population [60, 61]. Thus, governments should be involved in planning and implementing packages that improve the economic status of society, particularly women, in order to improve the adequacy of ANC and other maternal and child health services.

Women's educational status was associated with adequate ANC uptake, with women with a higher educational level having a higher likelihood of receiving adequate ANC visits than women with no education. Demographic analysis of Nigeria [29, 62], Zmabia [40], and Ethiopia [10] supported this finding. This could be because educated women are more likely to hear, comprehend, and understand information about the importance of ANC services via various channels (print and social media), and they are more likely to receive an adequate number of ANC. Maternal education can also improve healthcare literacy, making women more aware of danger signs and potentially able to identify pregnancy complications. Education, on the other hand, is known to increase women's autonomy in household decision-making and ability to decide about their own safety, which may result in compliance with the recommended number of ANC visits [35].

Similarly, the likelihood of receiving adequate ANC visits was higher among autonomous women. This finding was supported by studies conducted in Nigeria [63, 64], Bangladesh [65], Afghanistan [66], and Indonesia [67]. This could be because women's health decision-making autonomy prevents men or their male partners from exerting dominance and influence over women's decisions to seek ANC, resulting in a high number of ANC visits [68]. Restricting women's autonomy in making reproductive rights decisions may hinder the opportunity to obtain appropriate maternal health services [67]. There was broad agreement that a gender perspective should be taken into account in all development efforts, including sexual and reproductive health [69]. As a result, government bodies at the regional and national levels must work together to improve women's autonomy through interventions that increase their self-reliance and decision-making capacity.

The timing of ANC visits was a significant predictor of receipt of adequate ANC, with women who had their first ANC visit during the first trimester being 4.26 times more likely to receive adequate ANC. Studies conducted in Nigeria [29], Bangladesh [28, 70], Nepal [71] and Ethiopia [72] were in tandem with this finding. This could be because women who visit a health facility early in their pregnancy have enough time for consecutive visits, and the information they receive during the initial visit may facilitate continuity of care throughout the pregnancy, resulting in an adequate number of ANC visits [23]. Thus, encouraging mothers to

arrive early to begin ANC is critical to ensuring continuity of care, which opens the door to being informed about danger signs and complications during pregnancy.

The current study's findings were based on an analysis of the most recent large data sets covering 18 SSA countries, which were collected using standardized and validated data collection instruments and methodology, making the findings more generalizable. Besides, because of the clustering effect of DHS data, a multilevel-modeling approach was used in the analysis, providing disaggregated evidence on individual and community-level determinants for designing contextual interventions. Furthermore, the analysis considered both the timeliness and adequacy of ANC based on current WHO recommendations, and thus the findings could be used as input for SSA governments.

Even with the aforementioned benefits, the study has limitations. First, because the data were collected through retrospective interviews with a select group of women who had given birth within the previous five years, the findings may be prone to recall bias. Second, because the data is cross-sectional, the findings are subject to social desirability bias and fail to address the cause-effect relationship. It was also difficult to assess the relationship between other residual variables, such as cultural influence across countries.

## Conclusion

The findings revealed a low coverage of timely and adequate ANC visits in SSA countries, implying poor compliance with the WHO recommendation on ANC for positive pregnancy experiences. Planning status of the pregnancy, birth order, having a phone number, and enrollment in health insurance schemes were found to be significantly associated with the early initiation of ANC visits. On the other hand, Community-level poverty, the household wealth index, women's educational status, early initiation ANC visits, being autonomous in decision-making, and having a mobile phone were identified as significant predictors of adequate ANC service uptake. Governments and healthcare managers in SSA countries should leverage their efforts to prioritize and implement interventions that increase women's empowerment, particularly their autonomy in decision-making and economic capability, in order to improve their health-seeking behavior during pregnancy. More commitment is needed from governments to increase mobile phone distribution across countries, with a focus on households with childbearing women, and then work on integrating mHealth into their health system. Finally, efforts should be made to increase the coverage of health insurance schemes enrolment for their citizens.

## Acknowledgments

We would like to acknowledge the Demographic Health Survey program office for allowing us to access all the relevant DHS data for this study.

## Author Contributions

**Conceptualization:** Aklilu Habte.

**Data curation:** Aklilu Habte, Tamirat Melis.

**Formal analysis:** Aklilu Habte, Aiggan Tamene.

**Methodology:** Aklilu Habte, Aiggan Tamene, Tamirat Melis.

**Resources:** Aklilu Habte.

**Software:** Aklilu Habte.

**Supervision:** Aklilu Habte.

**Validation:** Aklilu Habte.

**Visualization:** Aklilu Habte, Aiggan Tamene.

**Writing – original draft:** Aklilu Habte, Aiggan Tamene, Tamirat Melis.

**Writing – review & editing:** Aklilu Habte, Aiggan Tamene.

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
