## [Decision Letter · Decision Letter 0]

6 Apr 2023

PONE-D-22-33293Compliance towards WHO recommendations on antenatal care for a positive pregnancy experience: Timeliness and adequacy of antenatal care visit in Sub-Saharan African countries: Evidence from the most recent standardized Demographic Health Survey dataPLOS ONE

Dear Dr. Habte Hailegebireal,

Thank you for submitting your manuscript to PLOS ONE. After careful consideration, we feel that it has merit but does not fully meet PLOS ONE’s publication criteria as it currently stands. Therefore, we invite you to submit a revised version of the manuscript that addresses the points raised during the review process.

We look forward to receiving your revised manuscript.

Kind regards,

Hiwot Yisak Dawid, MPH

Academic Editor

PLOS ONE

Journal Requirements:

2. Please check the tables "Table 1:  List of the individual- and community-level factors that affect the timing and adequacy of ANC, in SSA countries 2016-2021" "Table 1: Description of the SSA countries included in the analysis with their respective sample size, 2016-2021".

Additional Editor Comments:

The conclusion drawn seems shallow.Better to revise the conclusion. Basically the recommendations and conclusion has to be drawn from the main finding of the study and it has to be specific enough

work on modifying the language generally (spelling and grammar

Reviewers' comments:

Reviewer's Responses to Questions

**Comments to the Author**

1. Is the manuscript technically sound, and do the data support the conclusions?

Reviewer #1: Yes

2. Has the statistical analysis been performed appropriately and rigorously? 

Reviewer #1: Yes

3. Have the authors made all data underlying the findings in their manuscript fully available?

Reviewer #1: Yes

4. Is the manuscript presented in an intelligible fashion and written in standard English?

Reviewer #1: No

5. Review Comments to the Author

Reviewer #1: Dear Editorial Team and Authors: You asked me to review this study titled "Compliance towards WHO recommendations on antenatal care for a positive pregnancy experience: timeliness and adequacy of antenatal care visits in Sub-Saharan African countries: Evidence from the most recent standardized Demographic Health Survey data," and I am so grateful for the opportunity. The issue is intriguing. However, it comes with a lot of limitations. I will be glad to suggest the paper's publication if the author attempts to address the following topics. I am so very excited during reviewing this manuscript. Despite that I have attached some of the comments that should be addressed below. You can down load and try to correct carefully.

1. There are many grammatical errors in the manuscript, which is recommended to edit for the English language with regard to grammar, punctuation, spelling, line spacing, and clarity and needs correction

2. Try to avoid abbreviations / acronyms from abstract section. Before writing the full word, don’t use them also. E.g. DHS in abstract section.

3. What is the importance of dealing adequate and timely initiation of ANC rather than studying inadequate and delayed/late initiation of ANC? What it implies at the end?

4. The title should Contain as few words as much possible. But your title is broad and not easy to understand, so make it prices, short and clear.

5. There is an inconsistency of included data between title and method sessions; from 2016 to 2021 and from 2016 to 2020

6. The study was conducted in sub-Saharan African countries and there are more than 46 countries here, so why you were including only 18 SSA countries

7. Please don’t start a word with capital letter in the middle of a sentence. E.g. Community-level poverty, (at paragraph 3 line 3 of discussion section).

8. In fact, my big concern is what is worth this study more than other similar studies like a study conducted in Ethiopia and other SSA countries with the same issue. in addition, there is no implementation of the new WHO ANC recommendation launched in 2016 in our setup. So what is the relevance of this study?

9. You were selecting the final model based AIC, which was the fourth one with the lowest AIC.

Why you didn’t consider Deviance for selecting best model fit.

10. In the result part in Determinants of adequate ANC visits among women “the likelihood of receiving adequate ANC visits was 3,63 times higher” what does it mean? try to correct it.

11. What is the standard or bases for classifying ANC to adequate and in adequate especially in LMIC including Ethiopia the new ANC recommendation was not applied still now?

12. In your conclusion part Why you say low coverage of the two outcome of interest? What is the base for saying that?

6. PLOS authors have the option to publish the peer review history of their article (what does this mean?). If published, this will include your full peer review and any attached files.

Reviewer #1: No

---

## [Author Response · Author response to Decision Letter 0]

18 Apr 2023

The question raised by the editor was addressed and highlighted in the "data availability" section of the "revised manuscript with track changes".

---

## [Decision Letter · Decision Letter 1]

14 Nov 2023

Compliance towards WHO recommendations on antenatal care for a positive pregnancy experience: Timeliness and adequacy of antenatal care visit in Sub-Saharan African countries: Evidence from the most recent standardized Demographic Health Survey data

PONE-D-22-33293R1

Dear Dr. HABTE,

We’re pleased to inform you that your manuscript has been judged scientifically suitable for publication and will be formally accepted for publication once it meets all outstanding technical requirements.

Kind regards,

Rajesh Raushan, PhD

Academic Editor

PLOS ONE

Additional Editor Comments (optional):

None

Reviewers' comments:

Reviewer's Responses to Questions

**Comments to the Author**

1. If the authors have adequately addressed your comments raised in a previous round of review and you feel that this manuscript is now acceptable for publication, you may indicate that here to bypass the “Comments to the Author” section, enter your conflict of interest statement in the “Confidential to Editor” section, and submit your "Accept" recommendation.

Reviewer #1: All comments have been addressed

2. Is the manuscript technically sound, and do the data support the conclusions?

Reviewer #1: Yes

3. Has the statistical analysis been performed appropriately and rigorously? 

Reviewer #1: Yes

4. Have the authors made all data underlying the findings in their manuscript fully available?

Reviewer #1: Yes

5. Is the manuscript presented in an intelligible fashion and written in standard English?

Reviewer #1: Yes

6. Review Comments to the Author

Reviewer #1: (No Response)

7. PLOS authors have the option to publish the peer review history of their article (what does this mean?). If published, this will include your full peer review and any attached files.

Reviewer #1: No

---

## [Editor Report · Acceptance letter]

12 Jan 2024

PONE-D-22-33293R1 

PLOS ONE

Dear Dr. Habte, 

I'm pleased to inform you that your manuscript has been deemed suitable for publication in PLOS ONE. Congratulations! Your manuscript is now being handed over to our production team.

Kind regards, 

on behalf of

Dr. Rajesh Raushan 

Academic Editor

PLOS ONE